# Experimental Investigation of the Fluidization Reduction Characteristics of Iron Particles Coated with Carbon Powder under Pressurized Conditions

**DOI:** 10.3390/molecules25081810

**Published:** 2020-04-15

**Authors:** Qiyan Xu, Zhiping Li, Zhanghan Gu

**Affiliations:** 1School of Metallurgical Engineering, Anhui University of Technology, Ma Xiang Road, Ma’anshan 243002, China; 15005534845@163.com (Z.L.); 17755522068@163.com (Z.G.); 2The Key Laboratory of Functional Molecular Solids, Ministry of Education, Anhui Laboratory of Molecule-Based Materials, College of Chemistry and Materials Science, Anhui Normal University, Wuhu 241000, China

**Keywords:** coated carbon powder, pressure, sticking ratio, metallization rate, iron whiskers

## Abstract

The purpose of this study was to comprehensively analyze the effects of the carbon powder coating mass fraction, pressure, reduction temperature, reduction time, gas linear velocity, and particle size on fluidization reduction. Brazilian fine iron ore particles were the experimental object, and reduction experiments were performed under added carbon powder coating and pressure conditions. A six-factor, three-level orthogonal experiment method was used to obtain the optimal operating conditions and investigate the adhesion and inhibition mechanisms of fine iron ore during reduction. The experimental results show that with the addition of a carbon powder coating, an appropriate increase in pressure can increase the metallization rate, improve the fluidization state, and reduce the sticking ratio. The optimal operating conditions for pure hydrogen to reduce Brazilian fine iron ore was found to be a reduction temperature of 923–1023 K, the linear velocity of the reducing gas was 0.6 m/s, the reducing time was 30–50 min, the reducing pressure was 0.25 MPa, the mass content of the coated carbon powder was 2–6% (accounting for the mass of the mineral powder), and the particle size of the carbon powder was 4–7 µm. Iron whiskers cohesion and agglomeration were the main reasons for the adhesion of ore powder particles. It was found that carbon powder coating can effectively change the morphology of metal iron, as metal iron generates spherical particles around the carbon powder to improve the fluidization state.

## 1. Introduction 

Although blast furnace iron-making currently occupies a dominant position in the smelting of iron and steel, its reliance on metallurgical coke and emissions of pollutants, such as those generated during sintering and pelletizing, cause harm to humans and the environment [1,2,3]. To reduce the reliance of iron-making on metallurgical coke, the majority of metallurgical researchers have explored different types of non-blast furnace iron-making methods via long-term research [4]. The fluidized direct reduction process has the advantages of a large gas-solid interfacial contact area, uniform temperature and concentration, good heat and mass transfer conditions, and high operating efficiency, and meets both the metallurgical and environmental protection requirements. However, problems such as sticking or flow loss are prone to occur during the high temperature fluidization reduction process, thus reducing the reduction efficiency and hindering the continuous operation of the process. These problems have severely restricted the industrial application of fluidized iron-making [5,6,7,8]. 

To reduce the sticking ratio and increase the metallization rate, scholars have conducted a significant amount of research, primarily to inhibit the adhesion via the following strategies: (1) the improvement of the operating parameters, such as reducing the reduction temperature or increasing the air velocity [9,10]; (2) the addition of inert inhibitors such as silica and calcium oxide [11]; (3) the improvement of reduction equipment, the adoption of staged reduction, or the improvement of fluidized bed agitation equipment [12]. Mohannad and Maroufi [13,14] believed that the properties of the toner (the structure of the toner and the degree of graphitization) play an important role in the reduction kinetics. Fang and Li [15,16] proposed the use of a mixed fluidized bed method with ore and coal to carry out fluidization reduction experiments of iron ore powder, which reduced the adhesion ratio of the ore powder after reduction. Zhu and Shao [17,18,19] used different composition ratios of CO-H_2_ gas mixtures to reduce fine iron ore particles. It was found that when the gas mass fraction of carbon monoxide increased greatly, the surface of the ore powder was covered with a carbon layer, which could effectively prevent adhesion of iron ore powder. Summarizing previous studies, the addition of inert inhibitors is the most effective method for particle adhesion inhibition. Therefore, it is of great significance to further study the mechanism of adhesion inhibition due to carbon coatings. 

In this paper, Brazil iron ore powder is used as the experimental material to conduct iron ore powder reduction experiments in a high-pressure visible fluidized bed. First, four sets of preliminary experiments were conducted to analyze the effects of pressure and coated carbon powder on the fluidized reduction of Brazilian iron ore powder. Then we selected six influencing factors to design 27 sets of orthogonal experiments to determine the primary and secondary factors affecting the fluidization reduction effect of Brazilian iron ore powder through the experimental results, and obtain the best operating parameters through a comprehensive analysis. Finally, the mechanism of iron ore powder particle adhesion and the mechanism of coating carbon powder to inhibit iron ore powder adhesion are analyzed, which provide data and theoretical support for improving the fluidized ironmaking process. 

## 2. Methodology 

### 2.1. Experimental Materials 

In this experiment, Brazilian mineral powder with a particle size range of 0.15–0.18 mm was selected as the raw material (Table 1). Brazilian iron ore powder is mainly composed of hematite, T_Fe_ refers to the total iron content in iron ore powder. Carbon powder was selected as the coating agent in the experiments (Table 2). The particle sizes of the carbon powder were 4–7 µm, 48–75 µm, and 150–180 µm. 

### 2.2. Experimental Instruments and Methods 

A high-pressure transparent fluidized bed was selected for this experiment (Figure 1 and Figure 2). The reactor consisted of two layers of stainless steel tubes, and the inner tube was a fluidized bed area. The lower part is loaded with iron ore powder particles through a gas mixing hole. The reducing gas flows into the inner tube from the interlayer of the two layers of steel tubes and preheates it, and then flows into the inner tube. The flow rate of H_2_ and N_2_ is regulated by several groups of flow controllers, so that the gas flowing into the fluidized bed can reach the content composition and linear velocity required for each experiment. The linear velocity at the inlet end of the fluidized bed is controlled by regulating it using a gas mass flow controller, and the reduction temperature in the fluidized bed is measured by a thermocouple. Before the experiments, inert gas was introduced to check the air tightness of the device; after each experiment, inert gas was introduced to protect the reduction ore powder from any oxidation. The lamination difference Δ*P* of the bed was measured by a pressure gauge, and the state of fluidized reduction adhesion loss was judged through an observation window.

### 2.3. Experimental Design

Under the conditions of a reduction temperature of 1023 K, a gas linear velocity of 0.6 m/s, an ore particle size of 0.15–0.18 mm, and a reduction time of 50 min, 20 g of iron ore powder f was added to the tube or each experiment. Control experiments were performed under normal- and high-pressure conditions with fine iron ore particles with and without the addition of carbon powder in (Table 3). 

Before starting the experiment, a constant flow of N_2_ was introduced into the fluidized bed to exhaust the internal air, the air outlet valve was closed, the pressure of fluidized bed was increased to 0.5–0.6 MPa, and the air tightness of the device was checked. Then the fluidized bed starts to heat up. When the reduction temperature reached the design temperature (1023 K) the N_2_ valve was closed and the H_2_ valve was opened to begin the experiment. The test was then stopped after the set reduction time was reached. After the temperature of the furnace had cooled to room temperature, the reduced product in the tube was removed and stored in a sealed container. 

After cooling to room temperature, the bonded and unbonded mineral powder [20,21] in the fluidized bed were taken out, and the sticking ratio was calculated. The sticking ratio is the ratio of the bonded mass of the reduced powder to the total mass of the reduced powder. The smaller the sticking ratio, the better the fluidization state. The potassium dichromate volume method and the ferric chloride titration method were used to analyze the reduced samples to determine the metallic iron (*M*_Fe_) and full iron (*T*_Fe_) contents, and the metallization rate *η* was calculated (*η = M_Fe_ / T_Fe_*). Metallization rate refers to the ratio of the metal iron content to the total iron content. The metal iron should include zero-valent iron and the amount of iron combined with carbon. Total iron refers to the total iron content determined by chemical analysis of iron ore samples. The higher the metallization rate, the better the quality of the ore powder after reduction. The lower the sticking ratio, the better the fluidization state during the reduction process. Therefore, the metallization rate and the sticking ratio were selected as the indicators for determining the effect of fluidization reduction [22,23,24,25].

A comparative experimental study with and without the addition of carbon powder was performed at a temperature of 1023 K and a linear velocity of 0.6 m/s. In the Table 3, by comparing Scheme 1 with Scheme 2, it can be seen that the reduction pressure was 0.4 MPa, the mass fraction was 4%, and the particle size range of carbon powder was 48–75 µm. It was found that the sticking ratio was reduced by 8.92%, the fluidization effect was improved, and the metallization rate was reduced by 3.87%. By comparing Scheme 4 compared Scheme 3, it is evident that, after the addition of carbon powder with a particle size range of 48–75 of a mass fraction of 4% and at atmospheric pressure, the sticking ratio was reduced by 8.91%, the fluidization effect was improved, and the metallization rate was reduced by 3.93%. To sum up, under both normal- and high-pressure conditions, adding a certain particle size and proportion of carbon powder to iron ore powder was found to significantly reduce the sticking ratio and improve the fluidization effect. 

A comparative experimental study with high- and normal-pressure conditions was carried out at a temperature of 1023 K and a linear velocity of 0.6 m/s. In the Table 3, by comparing Scheme 1 with Scheme 3, it is clear that the reduction pressure increased from 0.1 to 0.4 MPa, the sticking ratio increased by 0.86%, and the metallization rate increased by 2.44%. In the Table 3, by comparing Scheme 4 with Scheme 2, it can be seen that by adding carbon powder with a particle size range of 48–75 µm at a mass fraction of 4%, the reduction pressure was increased from 0.1 to 0.4 MPa, the metallization rate was increased by 2.5%, and the sticking ratio was increased by 0.87%. In summary, under both the conditions of the addition and absence of carbon powder, an appropriate increase of the reduction pressure was found to improve the metallization rate. 

According to the results of the preliminary experiments, appropriately increasing the reduction pressure and adding carbon powder of a certain particle size to fine iron ore particles can help improve the metallization rate and the fluidization effect. 

Six factors that affect the sticking of Brazilian fine iron ore particles were investigated, namely the reduction temperature, the linear velocity of the reduction gas, the reduction time, the reduction pressure, the particle size of the toner, and the mass content of the toner. Additionally, three levels were considered for each factor. A, B, C, D, E, and F respectively represent the six factors in (Table 4), and the orthogonal experiment plan [26,27,28] is described in (Table 5).

The metallization rate and the sticking ratio were measured by the titration method. The bonding condition in the reduction process was judged by the pressure difference between the inlet and outlet gas. Fluidized reduction aims to achieve a high metallization rate without sticking. Therefore, a high metallization rate and a low sticking ratio were chosen as the optimal scheme.

## 3. Experimental Results and Discussion

### 3.1. Optimal Operating Parameters

The metallization ratios and sticking ratios as measured by the experiments are presented in Table 6, Table 7 and Table 8.

Comparing the extreme differences of the six factors, the optimal process parameters for the metallization rate were *A*_2_, *B*_2_, *C*_3_, *D*_2_, *F*_1_, and *E*_1_, i.e., a reduction temperature of 1023 K, a gas linear velocity of 0.6 m/s, a reduction time of 70 min, a reduction pressure of 0.25 MPa, an added carbon powder mass fraction of 2%, and an added carbon powder particle size of 4–7 μm. The main factor that affected the metallization rate was the temperature. The optimal process parameters for the sticking ratio were *F*_3_, *A*_1_, *E*_1_, *B*_2_, *D*_2_, and *C*_1_, i.e., an added carbon powder mass fraction of 6%, a reduction temperature of 973 K, an added carbon particle size of 4–7 µm, a gas linear velocity of 0.6 m/s, a reduction pressure of 0.25 MPa, and a reduction time of 30 min. The main factor that affected the sticking ratio was the mass fraction of added carbon powder.

The optimal solutions were obtained under given factors and levels; however, if a given level is not limited, it is possible to obtain a better experimental protocol. To obtain more accurate parameters, the selected factors and level ranges were appropriately adjusted to find new and better solutions. Taking the factor level as the abscissa and the average value of the experimental index *k_i_* as the ordinate, the relationship between the factor and the index is presented in Figure 3.

A comprehensive optimization plan can be obtained by the comprehensive balance method. The specific balance process is as follows:

*Factor A (Reduction temperature):* During the reduction process, the metallization rate gradually increased with the reduction reaction. When the reduction temperature reached a certain level, the metallization rate reached a maximum. As the reduction temperature increased, the metallization rate slightly decreased. Therefore, in terms of optimizing the metallization rate, level 2 was selected. During low-temperature reduction, the sticking ratio was maintained at a very low level, and the fluidization effect was good. As the temperature increased, the sticking ratio gradually increased, and the fluidization effect gradually deteriorated. Therefore, in terms of optimizing the sticking ratio, the reduction temperature was selected to be at level 1. Thus, overall, the temperature was selected to be between level 1 and level 2, i.e., 923–1023 K.

*Factor B (Linear velocity)*: Increasing the linear velocity of the gas has the effect of preventing the occurrence of adhesion loss and the production of fine iron ore particles with a higher metallization rate. However, in a cylindrical fluidized bed, there is an upper limit to the flow rate of the reducing gas. When the gas velocity in the fluidized bed exceeds the critical fluidization speed, a large number of air bubbles appear in the bed, and the air bubbles continue to rise. When they reach the surface of the bed, they will rupture and escape. During this process, many unreduced fine particles will be blown out of the fluidized bed due to the Yang Xi effect [29,30], causing a large amount of entrainment loss. According to the experimental results in Figure 3, level 2 was selected for the optimization of both the metallization rate and sticking ratio, i.e., the linear velocity of the reducing gas was 0.6 m/s.

*Factor C (Reduction time):* It can be seen from Figure 3 that as the reduction time increased, the metallization rate first increased rapidly, and then decreased gradually after 50 min of reduction, indicating that the reaction basically reached equilibrium after 50 min. Therefore, level 2 was chosen for the optimization of the metallization rate, i.e., the reduction time was chosen to be 50 min. Because the coated carbon powder inhibited the fine iron ore particles from sticking, which caused the fine iron ore particles to remain fluidized for a long time during the reduction process, it can be seen from Figure 3 that the sticking ratio increased slowly with time. Level 1 was therefore selected for the optimization of the sticking ratio, i.e., a reduction time of 30 min was selected. Thus, overall, the reduction time was chosen to be between 30 and 50 min.

*Factor D**(Reduction pressure**):* With the increase of reduction pressure, the gas phase density increases, as does the gas-solid contact, which accelerates the gas-solid reaction rate and increases the metallization rate [31,32,33]. However, as the reduction pressure increases, the speed of the reduction gas increases, which causes the reduction gas to be discharged from the fluidized bed container before it reaches the set temperature. The effective gas-solid reduction temperature decreases, as do the reduction effect, the metallization rate, and the effective utilization of gas, resulting in a reduction in the metallization rate. Therefore, level 2 was selected for the optimization of the metallization rate and effective gas utilization. With the progression of the reaction, the precipitated iron whiskers or iron atoms co-aggregate with each other, resulting in increases in the size and gravity of the particles. When the gravity and the cohesive force are greater than the drag force of the gas on the particles, the particles will stick together, and the sticking ratio will increase. Therefore, level 2 was also selected for the optimization of the sticking ratio. Thus, overall, the reduction pressure was chosen to be 0.25 MPa.

*Factor E**(Carbon powder particle size**):* By forming a carbon coating on the outer surface of the mineral powder particles, the direct contact of the metal iron on the surface of the particles can be isolated to achieve the purpose of restraining the particles from adhering to each other. However, the specific effect and related mechanism of the particle size of the toner particles on the adhesion/loss are also related to the particle size of the minerals, the linear velocity of the gas, and the mass content of the toner particles. From the experimental results, it can be seen that the particle size of the coated carbon particles was relatively small, and the metallization rate reached a maximum of 78.07%. Level 1 was therefore selected for the optimization of the metallization rate, i.e., the particle size of the added carbon powder was chosen to be 4–7 μm. The finer the carbon powder particles, the easier it is to adsorb on the surface of rough Brazil mineral powder, which changes the morphology of the precipitated metallic iron and suppresses adhesion. In terms of the optimization of the adhesion ratio, level 1 was also selected. Thus, overall, the particle size of the toner particles was selected to be between 4 and 7 μm.

*Factor F (Mass content of toner):* As the mass content of toner particles increased, the metallization rate remained at a high level, and the sticking ratio decreased significantly. This is because the coating of carbon powder blocked the hooking of iron whiskers or the agglomeration of iron atoms, effectively improving the metallization rate. However, as the content of carbon powder particles increased, the amount of precipitated graphite also increased, and the newly precipitated iron whiskers or iron atoms were effectively coated. Additionally, the unreacted ore powder particles were also covered to a certain extent, which reduced the effective contact area between the reducing gas and the ore powder particles, and also reduced the reaction efficiency. This is also the reason for the downward trend after the maximum metallization rate occurred. Therefore, for the optimization of the metallization rate and the effective use of gas, level 1 was selected, and for the optimization of the sticking ratio, level 3 was selected. Thus, overall, the toner content was chosen to be between 2% and 6%. Comprehensively, the best operating parameters were selected as follows: a temperature of 923–1023 K, a reduction gas velocity of 0.6 m/s, a reduction time of 30–50 min, a reduction pressure of 0.25 MPa, a toner particle size of 4–7 μm, and a toner content of 2–6%.

### 3.2. Bonding Mechanism of Fine Iron Ore Particles

The reaction of reducing iron oxide with a reducing gas (*H_2_*) in this experiment is called the direct reduction method. In this method, the reducing gas doubles as a reducing agent and a heat carrier, and is carried out in the middle- and low-temperature region of the furnace. Because the reduction of Fe_2_O_3_, Fe_3_O_4_, and FeO are all reversible reactions, excess reducing agents are required to ensure the smooth progress of the reaction.

With H_2_ as a reducing agent, the reactions that occur during the reduction of iron ore are as follows [34,35].
3Fe_2_O_3(s)_ + H_2(g)_ = 2Fe_3_O_4(s)_ + H_2_O_(g)_(1)
Fe_3_O_4(s)_ + H_2(g)_ = 3FeO_(s)_ + H_2_O_(g)_(2)
FeO_(s)_ + H_2(s)_ = Fe(s) + H_2_O_(g)_(3)
Fe_3_O_4(s)_ + 4H_2(g)_ = 3Fe(s) + 4H_2_O_(g)_(4)

As is clear from the Figure 4, the free energy of reaction 1 is less than zero, and a forward reaction is most likely to occur. It can also be reduced to Fe_3_O_4_ under low reducing atmosphere conditions. If the temperature is less than 843 K, Fe_2_O_3_ is reduced to Fe_3_O_4_; if the temperature is greater than 843 K, Fe_2_O_3_ is quickly reduced to Fe_3_O_4_, then to FeO, and finally to metallic iron, and a small part of the Fe_3_O_4_ is directly reduced to metallic iron.

Assuming that the iron ore reaction in the experiment follows an unreacted shrinking core model of iron oxide particle reduction, a differential equation for iron ore reduction kinetics can be established based on the model. The reduction product gas (H_2_O) of the reduction of Fe_2_O_3_ particles in a reducing gas atmosphere (H_2_) is taken as an example. At temperatures greater than 843 K, iron oxides are gradually reduced.
(5)$$Fe_{2}O_{3}\to Fe_{3}O_{4}\to FeO\to Fe$$

As the reduction reaction progresses, a metallic Fe product layer is gradually formed, and the core radius of unreacted Fe_2_O_3_ is gradually reduced in (Figure 5). The restoration process can be divided into the following steps:

(1)The reducing gas (H_2_) diffuses through the gas film to the surface of the solid product layer (Fe). This process is called external diffusion;(2)The reducing gas (H_2_) diffuses to the reaction interface through the product layer, and iron ions diffuse into the interior through the product layer. This process is called internal diffusion(3)The adsorption of the reducing gas (H_2_), the interface chemical reaction, and the desorption of the oxidizing gas have the characteristics of adsorption autocatalysis, i.e., their rate has a phenomenon of small to large autocatalysis.

According to the unreacted shrinking core model of gas-solid reaction, the H_2_ gas reduction reaction of Fe_2_O_3_ particles can be written as:H_2_ + Fe_x_O = xFe + H_2_O(6)

If the resistance of external diffusion, internal diffusion, and interface chemical reaction during the reaction are not ignored, the reaction kinetic equation should consider the influences of these three factors on the rate simultaneously. For spherical particles, the following equation can be obtained [36,37]:(7)$$\begin{array}{c} t=\frac{r_{0}\rho_{Fe_{x}o}}{3k_{g}C_{H_{2}}M_{Fe_{x}o}}R_{s}+\frac{r_{0}^{2}\rho_{Fe_{x}o}}{6D_{eff}C_{H_{2}}M_{Fe_{x}o}}\left[1+2\left(1-R_{s}\right)-3{\left(1-R_{s}\right)}^{\frac{2}{3}}\right]\\ +\frac{r_{0}\rho_{Fe_{x}o}}{k_{rea}C_{H_{2}}M_{Fe_{x}o}}{\left[1-\left(1-R_{s}\right)\right]}^{\frac{1}{3}} \end{array}$$
where *t* is the time required to reach reduction degree, *min*; $r_{0}$ is the particle radius of $Fe_{x}o$, *m*; $\rho_{Fe_{x}o}$ is the particle density of $Fe_{x}o\;$, *kg/m^3^*; $k_{g}$ is the gas boundary layer mass transfer coefficient; $C_{H_{2}}$ is the *H_2_* gas concentration in the gas phase, 0.089 *kg/m^3^*; $M_{Fe_{x}o}$ is the Molar mass of $Fe_{x}o\;$, *g/mol*; $R_{s}$ is the degree of reduction when particles are bonded; $D_{eff}$ is the effective diffusion coefficient of *H_2_* gas; $k_{rea}$ is the chemical reaction rate constant; $t_{f}$ is the fluidization time, *min*.

An H_2_ reducing gas flow with a linear velocity of 0.6 m/s passes 20 g of Brazilian fine iron ore particles for fluidized reduction. The effect of different reduction temperatures (T = 973 K, 1023 K, 1073 K, 1123 K, 1173 K) on the average pressure difference was studied. Figure 6 presents the change of the average bed pressure during the fluidized reduction of Brazilian fine iron ore particles at different temperatures. It can be seen that at temperatures greater than 950 K, the average pressure difference of the bed decreased extremely rapidly with the increase of temperature. If the pressure difference is consistent at the beginning of the reduction, it can be inferred that the time of the bed pressure drop over the same reduction time gradually increases with the increase of temperature. It is evident that the fluidization time is significantly affected by the reduction temperature; the higher the reduction temperature, the smaller the average pressure difference, the shorter the fluidization time, and the worse the fluidity of particles.

During the fluidization reduction process, due to the reduction reaction, the phases of the Brazilian fine iron ore particles changed, and the change of the fluidization characteristics of the particles was closely related to the change of their phases. Figure 7 presents the relationship between the metallization rate and temperature of particles after reduction. The metallization rates of Fe_2_O_3_ particles after reduction at 873 K, 923 K, 973 K, 1023 K, 1073 K, 1123 K, and 1173 K were respectively 55.12%, 67.12%, 79.86%, 87.05%, 88.80%, 88.96%, and 86.81%. It is clear that during the reduction process of particles under the experimental conditions, the fluidization phase of particles was the evolution from Fe_2_O_3_ to Fe_3_O_4_, FeO, and metallic iron. Because the reduction temperature can also change the reduction kinetics of particles, the effect of the reduction temperature on the reduction kinetics of particles was analyzed.

In the gas-solid unreacted shrinking core model, changes in temperature will affect gas diffusion and chemical reaction rate constants, i.e., *k_g_*, *D_eff_*, and *k_rea_*. The empirical formula for laminar forced convective fluid passing through the surface of a sphere is as follows:(8)$$\frac{k_{g}d}{D}\approx 0.6Re^{\frac{1}{2}}Sc^{\frac{1}{3}}$$
where d is the Particle diameter, *m*; D is the gas reaction diffusion coefficient, *m^2^/s*; Re is the Reynolds number; Sc is the Schmidt number.

So one can get:(9)$$k_{g}\approx \frac{0.6Re^{\frac{1}{2}}Sc^{\frac{1}{3}}D}{d}$$

The relationship between diffusion coefficient (*D*), reaction rate constant (*k_rea_*) and temperature (*T*) is as follows:(10)$$D=D_{0}e^{-\frac{Q_{1}}{RT}}$$
(11)$$\;\;k_{rea}=Ae^{-\frac{Q_{2}}{RT}}\;\;$$

Substituting the above formula into Equation (7), we get:(12)$$\begin{array}{cc} T\approx \frac{r_{0}\rho_{Fe_{x}o}}{3C_{H_{2}}M_{Fe_{x}o}}R_{s} & \times \frac{d}{0.6Re^{\frac{1}{2}}Sc^{\frac{1}{3}}D_{0}}\times e^{\frac{Q_{1}}{RT}}\\ & +\frac{r_{0}^{2}\rho_{Fe_{x}o}}{6D_{0}C_{H_{2}}M_{Fe_{x}o}}\left[1+2\left(1-R_{s}\right)-3{\left(1-R_{s}\right)}^{\frac{2}{3}}\right]\times e^{\frac{Q_{1}}{RT}}\\ & +\frac{r_{0}\rho_{Fe_{x}o}}{AC_{H_{2}}M_{Fe_{x}o}}{\left[1-\left(1-R_{s}\right)\right]}^{\frac{1}{3}}\times e^{\frac{Q_{2}}{RT}} \end{array}$$

The following calculation ignores the activation energy subscript (*Q = Q_1_ = Q_2_*), then one can get:(13)$$\begin{array}{c} T=\{\frac{r_{0}\rho_{Fe_{x}o}}{3C_{H_{2}}M_{Fe_{x}o}}R_{s}\times \frac{d}{0.6Re^{\frac{1}{2}}Sc^{\frac{1}{3}}D_{0}}+\frac{r_{0}^{2}\rho_{Fe_{x}o}}{6D_{0}C_{H_{2}}M_{Fe_{x}o}}\left[1+2\left(1-R_{s}\right)-3{\left(1-R_{s}\right)}^{\frac{2}{3}}\right]\\ +\frac{r_{0}\rho_{Fe_{x}o}}{AC_{H_{2}}M_{Fe_{x}o}}{\left[1-\left(1-R_{s}\right)\right]}^{\frac{1}{3}}\}\times e^{\frac{Q}{RT}} \end{array}$$

In the temperature range of 973 ~ 1173 K, the density and viscosity of the gas change little with time, which is ignored:(14)$$\begin{array}{c} f_{T}(R_{s})=\frac{r_{0}\rho_{Fe_{x}o}}{3C_{H_{2}}M_{Fe_{x}o}}R_{s}\times \frac{d}{0.6Re^{\frac{1}{2}}Sc^{\frac{1}{3}}D_{0}}+\frac{r_{0}^{2}\rho_{Fe_{x}o}}{6D_{0}C_{H_{2}}M_{Fe_{x}o}}\left[1+2\left(1-R_{s}\right)-3{\left(1-R_{s}\right)}^{\frac{2}{3}}\right]\\ +\frac{r_{0}\rho_{Fe_{x}o}}{AC_{H_{2}}M_{Fe_{x}o}}{\left[1-\left(1-R_{s}\right)\right]}^{\frac{1}{3}} \end{array}$$
and letting *t_f_* ≈ *t*, so:(15)$$t_{f}\approx f_{T}(R_{s})e^{\frac{Q}{RT}}$$

Logarithms on both sides can be obtained:(16)$$\ln t_{f}\approx \ln\left[f_{T}(R_{s})\right]+\frac{Q}{RT}$$

The results reveal that the natural logarithm of the fluidization time and the inverse of the reduction temperature have a good linear relationship, indicating that the change of the fluidization time with the reduction temperature also conforms to the reduction kinetics equation. The diffusion coefficient of the gas and the rate constant of the reaction determine the resistance of each reaction step, and their natural logarithms increase linearly with the inverse of temperature. The higher the temperature, the smaller the resistance of each reaction step, and the faster the particle phase evolution rate; thus, the fluidization time is shortened. The results show that the fine iron ore was rapidly reduced to metallic iron under high-temperature conditions, and the iron and gold on the surface of the ore powder were linked to each other, resulting in a worsened fluidization state and shorter fluidization time.

### 3.3. Mechanism of the Inhibition of Mineral Powder Bonding by Carbon Coating

During the carbon coating reduction process, carbon participates in the reaction to reduce iron oxide. The chemical reaction is as follows:
(17)$${\;\mathrm{Fe}}_{x}O_{y(s)}+{\mathrm{yC}}_{\left(s\right)}={\mathrm{xFe}}_{\left(s\right)}+{\mathrm{yCO}}_{\left(g\right)}$$
(18)$${\;\mathrm{Fe}}_{x}O_{y\left(s\right)}+\frac{y}{2}C_{\left(s\right)}={\mathrm{xFe}}_{\left(s\right)}+\frac{y}{2}{\mathrm{CO}}_{2\left(g\right)}$$
(19)$${\;C}_{\left(s\right)}+{\mathrm{CO}}_{2\left(g\right)}=2{\mathrm{CO}}_{\left(g\right)}$$
(20)$${\;\mathrm{Fe}}_{x}O_{y\left(s\right)}+{\mathrm{yCO}}_{\left(g\right)}={\mathrm{xFe}}_{\left(s\right)}+{\mathrm{yCO}}_{2\left(g\right)}$$

The reduction is also enhanced by iron oxide reduction by carbon. This reduction occurs by direct solid-solid reaction which produces carbon monoxide and carbon dioxide. Carbon dioxide again attacks carbon to form carbon monoxide (Boudouard reaction) and to form very reactive porous carbon [38]. Porous carbon enhances the reduction of iron oxide [39], while carbon monoxide is very good reductant for most metal oxide [40]. Therefore, the addition of carbon powder can improve the reduction effect of iron oxide.

Based on the results of several groups of comparative experiments, the mechanism of the adhesion of fine iron ore particles and the mechanism of the inhibition of the adhesion of fine iron ore particles by carbon coating were analyzed from the microscopic point of view. EDS analysis was carried out for the acicular and globular products on the surface of ore powder after reduction, and the results are presented in Figure 8 and Figure 9.

It is clear from Figure 8 that under the uncoated carbon powder conditions, there were slender hooks in the bonded area of the Brazilian ore powder, which were determined to be metal iron via energy spectrum analysis. When the metal was above the Tammann temperature [41,42], its lattice was no longer stable, the solid unit became active, and the metal iron atoms or iron whiskers began to diffuse, resulting in a solid bridging force between particles. This ultimately resulted in the bonding of particles above the Taman temperature.

Figure 8 reveals that after the addition of carbon powder coating, a large number of small balls with compact diameters of less than 1 μm were formed on the surfaces of the Brazilian mineral powder, and were determined to be metallic iron via energy spectrum analysis. As compared with the uncoated carbon powder, the coated carbon powder easily allowed the precipitated metal iron to grow around the carbon powder and form nanoscale-sized metal balls, which could inhibit the formation of iron whiskers, reduce the bond force of iron powder, reduce the sticking ratio of the reduced ore powder, and improve the fluidization reduction state.

## 4. Conclusions

The conclusions arising from this work can be summarized as follows:(1)Under carbon powder coating conditions, the metallization rate can be increased, the fluidization state can be improved, and the adhesion ratio can be reduced by properly increasing the pressure;(2)The best operating parameters of the coated toner were as follows. The temperature was selected to be 923–1023 K, the reduction gas velocity was selected to be 0.6 m/s, the reduction time was selected to be 30–50 min, the reduction pressure was selected to be 0.25 MPa, the particle size of the toner particles was selected to be 4–7 μm, and the toner content was selected to be 2–6%;(3)The agglomeration of iron whiskers is the main reason for the adhesion of iron powder particles. The carbon powder coating was found to effectively change the morphology of iron metal, and the iron metal formed spherical particles around the carbon powder to improve the fluidization state.

In the future, two experimental directions are planned. On the one hand, the fluidization experiments will be continued. A flue gas analyzer will be installed at the gas outlet to determine the degree of reduction of iron ore powder in the fluidized bed by analyzing the composition of the exhaust gas. On the other hand, the reduced iron ore powder will be subjected to hot briquetting treatment, and then put into an electric furnace as a steelmaking raw material to smelt special steel.

## Figures and Tables

**Figure 1 molecules-25-01810-f001:**
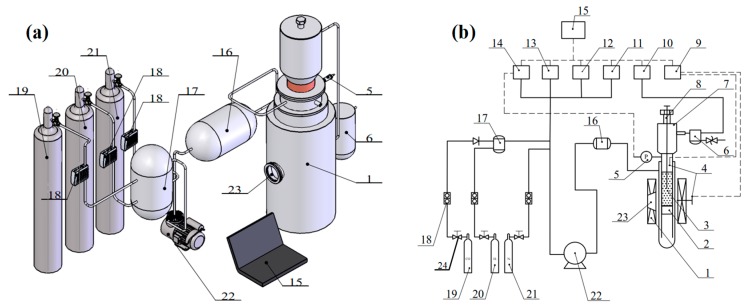
Experimental setup: (**a**) Fluidized bed reactor (**b**) The flow chart. 1. Gasmixing & preheating chamber 2. Gas mixing hole 3. Fluidized bed 4. K-type thermal couple 5. Pressure sensor 6. Gravity filter 7. Feeding and sampling port 8. Pressure seal cap 9. Temperature change recorder 10. Gas analyzer 11. H_2_ gas analysis recorder 12. CO_2_ gas analysis recorder 13. CO gas analysis recorder 14. Pressure change analysis recorder 15. Computer 16. Gas dryer 17. Gasholder 18. Pressure display 19. Gas mass flowmeter 20. CO/CO_2_ gas cylinders 21. H_2_ gas cylinders 22. N_2_ gas cylinders 23. Booster pump 24. Gas valve.

**Figure 2 molecules-25-01810-f002:**
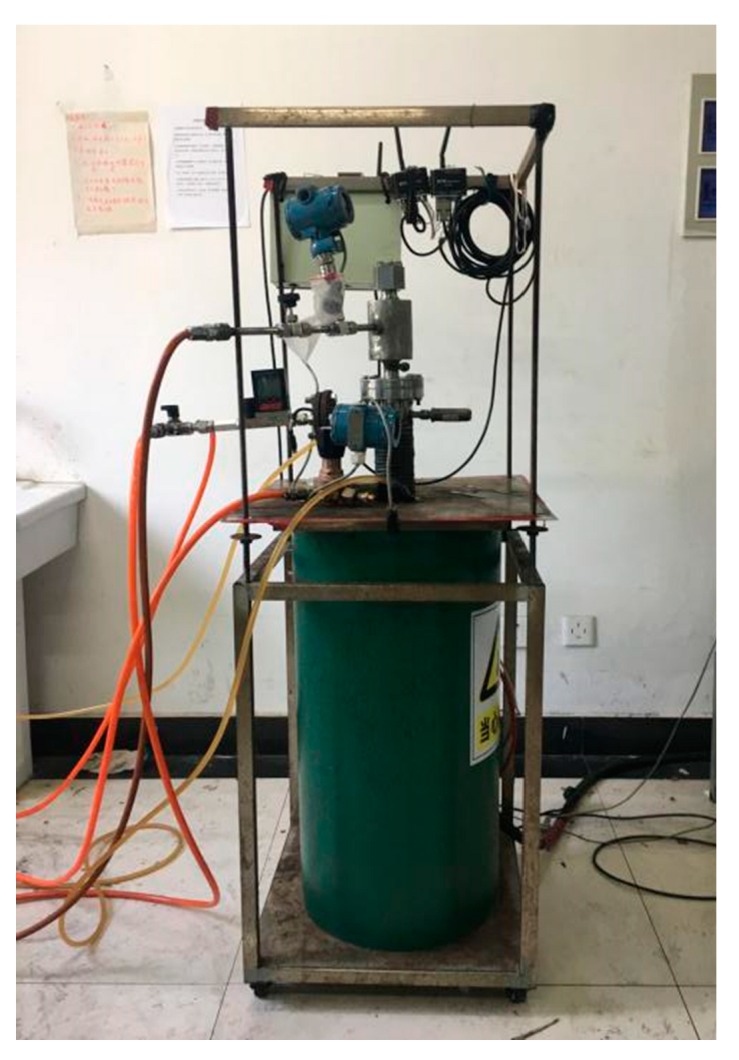
Experimental device.

**Figure 3 molecules-25-01810-f003:**
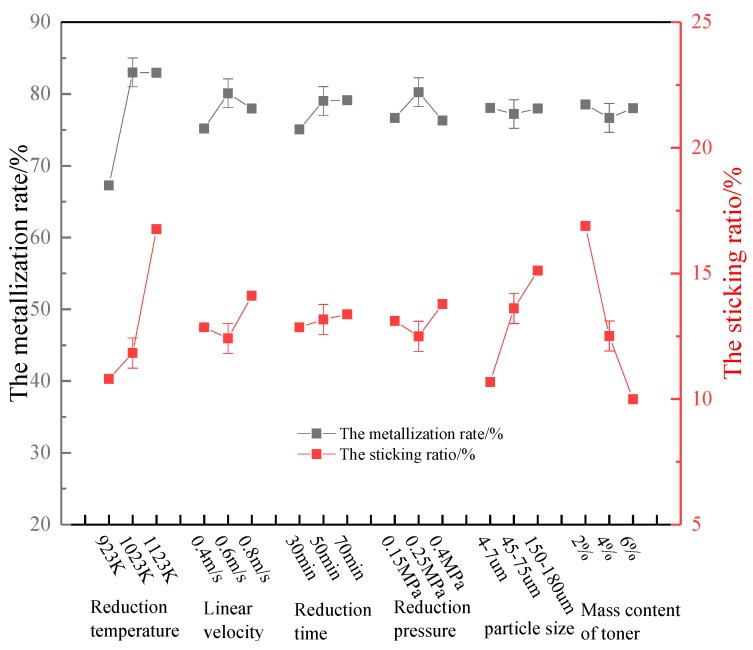
Relationship between metallization rate and adhesion ratio at different factor levels.

**Figure 4 molecules-25-01810-f004:**
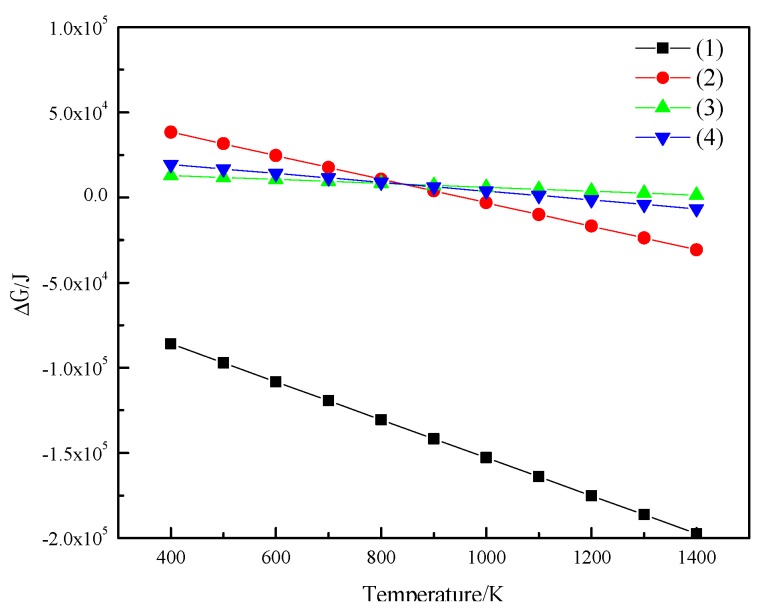
Gibbs free energy of H_2_ reduction of iron oxide.

**Figure 5 molecules-25-01810-f005:**
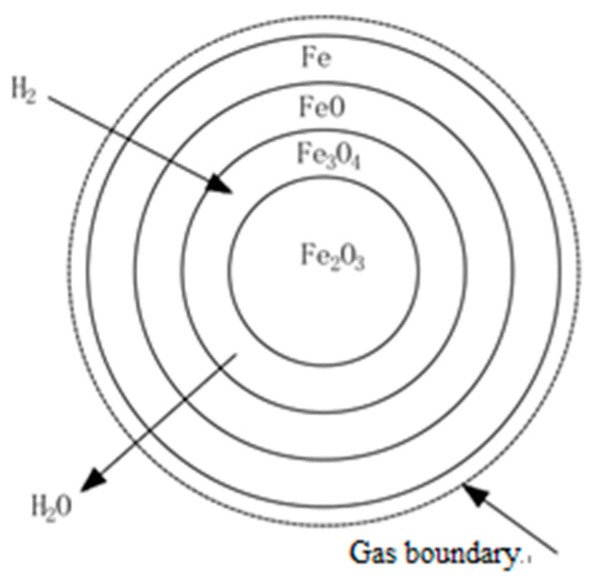
Unreacted shrinking core model.

**Figure 6 molecules-25-01810-f006:**
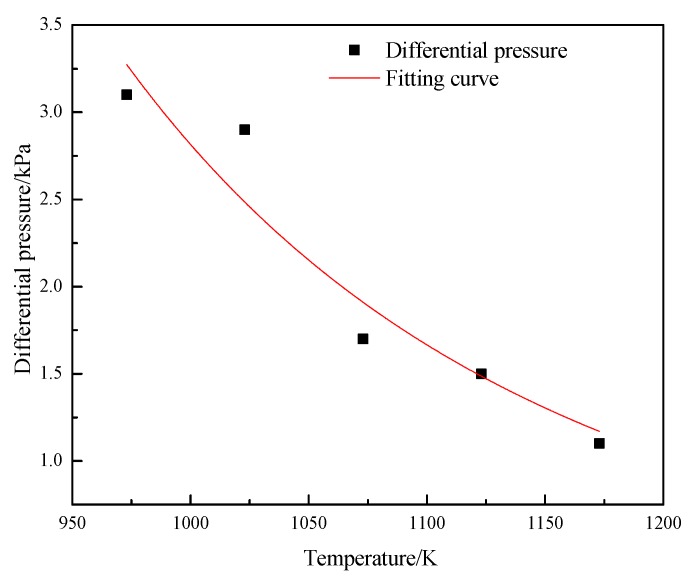
Change of average pressure difference with reduction temperature.

**Figure 7 molecules-25-01810-f007:**
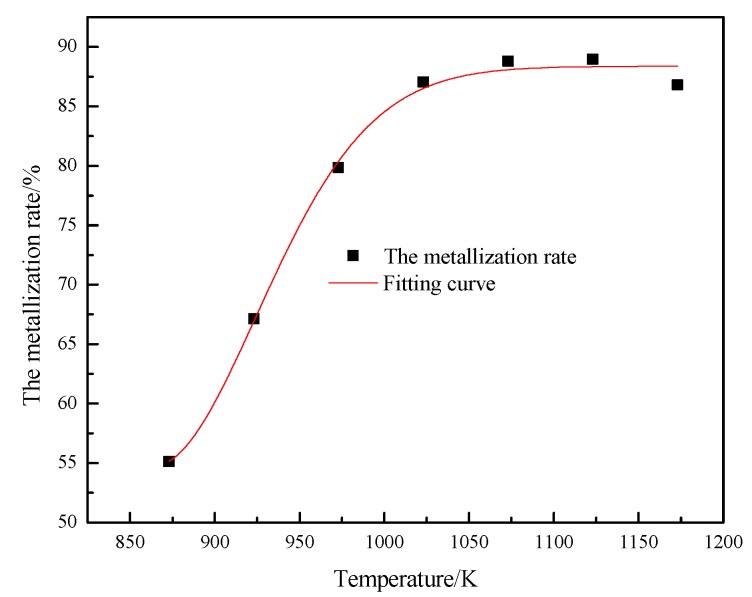
Metallization rate changes with temperature.

**Figure 8 molecules-25-01810-f008:**
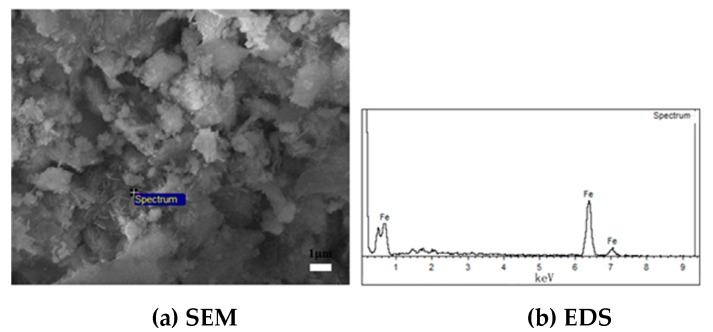
SEM analysis of Brazilian mineral powder without carbon powder coating.

**Figure 9 molecules-25-01810-f009:**
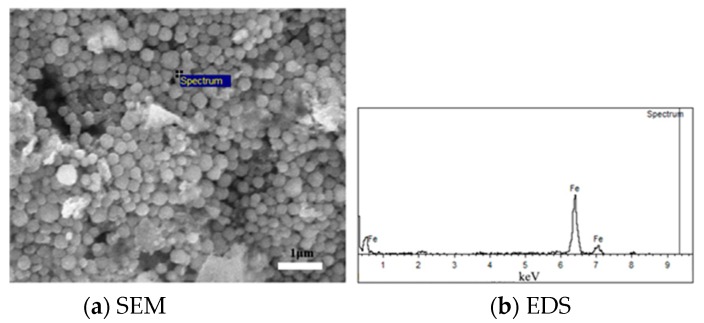
SEM analysis of Brazilian mineral powder under the condition of carbon powder coating.

**Table 1 molecules-25-01810-t001:** Chemical composition of Brazilian mineral powder (mass %).

T_Fe_	FeO	CaO	MgO	SiO_2_	P	MnO	Al_2_O_3_
52.31	0.88	0.16	<0.05	22.15	0.043	0.195	0.88

**Table 2 molecules-25-01810-t002:** Chemical composition of toner (mass %).

Fixed Carbon	Ash	Volatile Matter	Phosphorus Content	Moisture
>82	<15	<3	<0.015	<5

**Table 3 molecules-25-01810-t003:** Pre-experimental schemes and results.

Scheme	Reduction Pressure/MPa	Particle Size/µm	Toner Content/%	Metallization Rate/%	Sticking Ratio/%
1	0.4	0	0	85.00	18.35
2	0.4	48–75	4	81.13	9.43
3	0.1	0	0	82.56	19.21
4	0.1	48–75	4	78.63	10.30

**Table 4 molecules-25-01810-t004:** Level of each factor.

A Reduction Temperature/K			B Gas Linear Velocity/m/s			C Reduction Time/min			D Reduction Pressure/MPa			E Particle Size/µm			F Toner Content/%		
A_1_	A_2_	A_3_	B_1_	B_2_	B_3_	C_1_	C_2_	C_3_	D_1_	D_2_	D_3_	E_1_	E_2_	E_3_	F_1_	F_2_	F_3_
923	1023	1123	0.4	0.6	0.8	30	50	70	0.1	0.25	0.4	4–7	48–75	150–180	2	4	6

**Table 5 molecules-25-01810-t005:** Orthogonal test scheme.

	Reduction Temperature/K	Gas linear Velocity/m/s	Reduction Time/min	Reduction Pressure/MPa	Particle Size /µm	Toner Content/%
**1**	923	0.4	30	0.1	4–7	2
**2**	923	0.4	30	0.1	48–75	4
**3**	923	0.4	30	0.1	150–180	6
**4**	923	0.6	50	0.25	4–7	2
**5**	923	0.6	50	0.25	48–75	4
**6**	923	0.6	50	0.25	150–180	6
**7**	923	0.8	70	0.4	4–7	2
**8**	923	0.8	70	0.4	48–75	4
**9**	923	0.8	70	0.4	150–180	6
**10**	1023	0.4	50	0.4	4–7	4
**11**	1023	0.4	50	0.4	48–75	6
**12**	1023	0.4	50	0.4	150–180	2
**13**	1023	0.6	70	0.1	4–7	4
**14**	1023	0.6	70	0.1	48–75	6
**15**	1023	0.6	70	0.1	150–180	2
**16**	1023	0.8	30	0.25	4–7	4
**17**	1023	0.8	30	0.25	48–75	6
**18**	1023	0.8	30	0.25	150–180	2
**19**	1123	0.4	70	0.25	4–7	6
**20**	1123	0.4	70	0.25	48–75	2
**21**	1123	0.4	70	0.25	150–180	4
**22**	1123	0.6	30	0.4	4–7	6
**23**	1123	0.6	30	0.4	48–75	2
**24**	1123	0.6	30	0.4	150–180	4
**25**	1123	0.8	50	0.1	4–7	6
**26**	1123	0.8	50	0.1	48–75	2
**27**	1123	0.8	50	0.1	150–180	4

**Table 6 molecules-25-01810-t006:** Experimental scheme and results.

	Reduction Temperature/K	Gas linear Velocity/m/s	Reduction Time/min	Reduction Pressure/MPa	Particle Size /µm	Toner Content/%	Metallization Rate/%	Sticking Ratio/%
**1**	A_1_	B_1_	C_1_	D_1_	E_1_	F_1_	65.13	13.48
**2**	A_1_	B_1_	C_1_	D_1_	E_2_	F_2_	59.72	10.70
**3**	A_1_	B_1_	C_1_	D_1_	E_3_	F_3_	57.96	6.54
**4**	A_1_	B_2_	C_2_	D_2_	E_1_	F_1_	70.42	7.86
**5**	A_1_	B_2_	C_2_	D_2_	E_2_	F_2_	72.88	8.78
**6**	A_1_	B_2_	C_2_	D_2_	E_3_	F_3_	76.94	11.78
**7**	A_1_	B_3_	C_3_	D_3_	E_1_	F_1_	67.69	15.65
**8**	A_1_	B_3_	C_3_	D_3_	E_2_	F_2_	65.11	10.90
**9**	A_1_	B_3_	C_3_	D_3_	E_3_	F_3_	69.47	11.47
**10**	A_2_	B_1_	C_2_	D_3_	E_1_	F_2_	83.96	8.03
**11**	A_2_	B_1_	C_2_	D_3_	E_2_	F_3_	75.78	6.41
**12**	A_2_	B_1_	C_2_	D_3_	E_3_	F_1_	81.06	22.32
**13**	A_2_	B_2_	C_3_	D_1_	E_1_	F_2_	84.35	12.06
**14**	A_2_	B_2_	C_3_	D_1_	E_2_	F_3_	88.54	10.70
**15**	A_2_	B_2_	C_3_	D_1_	E_3_	F_1_	84.19	11.26
**16**	A_2_	B_3_	C_1_	D_2_	E_1_	F_2_	79.56	7.03
**17**	A_2_	B_3_	C_1_	D_2_	E_2_	F_3_	82.09	11.08
**18**	A_2_	B_3_	C_1_	D_2_	E_3_	F_1_	87.61	17.57
**19**	A_3_	B_1_	C_3_	D_2_	E_1_	F_3_	84.72	13.38
**20**	A_3_	B_1_	C_3_	D_2_	E_2_	F_1_	83.75	18.83
**21**	A_3_	B_1_	C_3_	D_2_	E_3_	F_2_	84.46	16.14
**22**	A_3_	B_2_	C_1_	D_3_	E_1_	F_3_	82.68	13.45
**23**	A_3_	B_2_	C_1_	D_3_	E_2_	F_1_	81.38	15.33
**24**	A_3_	B_2_	C_1_	D_3_	E_3_	F_2_	79.58	20.49
**25**	A_3_	B_3_	C_2_	D_1_	E_1_	F_3_	84.08	5.13
**26**	A_3_	B_3_	C_2_	D_1_	E_2_	F_1_	85.74	29.72
**27**	A_3_	B_3_	C_2_	D_1_	E_3_	F_2_	80.41	18.44

**Table 7 molecules-25-01810-t007:** Analysis of metallization rate results.

Factor	Indicators	A	B	C	D	E	F
**Metallization rate/%**	K_1_	605.32	676.54	675.71	690.12	702.59	706.97
	K_2_	747.14	720.96	711.27	722.43	694.99	690.03
	K_3_	746.80	701.76	712.28	686.71	701.68	702.26
	k_1_	67.26	75.17	75.08	76.68	78.07	78.55
	k_2_	83.02	80.11	79.03	80.27	77.22	76.67
	k_3_	82.98	77.97	79.14	76.30	77.96	78.03
	R	15.72	4.94	3.95	3.59	0.84	1.88
		Primary and secondary factors ABCDFE					
		Optimization scheme A_2_B_2_C_3_D_2_F_1_E_1_					

**Table 8 molecules-25-01810-t008:** Analysis of sticking ratio results.

Factor	Indicators	A	B	C	D	E	F
**Sticking mass percentage/%**	K_1_	97.16	115.67	115.67	118.03	96.07	152.02
	K_2_	106.46	111.71	118.47	112.45	122.45	112.57
	K_3_	150.91	126.99	120.39	124.05	136.01	89.94
	k_1_	10.80	12.85	12.85	13.11	10.67	16.89
	k_2_	11.83	12.41	13.16	12.49	13.61	12.51
	k_3_	16.77	14.11	13.38	13.78	15.11	9.99
	R	5.97	1.70	0.52	1.29	4.44	6.90
		Primary and secondary factors FAEBDC					
		Optimization scheme F_3_A_1_E_1_B_2_D_2_C_1_					

Note: K_i_ represents the sum of the test results for the corresponding level number in each column; k_i_ = K_i/s_ where s is the number of times the level appears on any column, s = 9 in this table; the range R indicates that it affects the Brazilian mine the primary and secondary of powder reduction are the maximum value of each factor level minus the minimum value, R = k_max_ − k_min_. The larger the R value, the more obvious the reduction effect of this factor on Brazilian mines.

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
