# Peer review of "Experimental Investigation of the Fluidization Reduction Characteristics of Iron Particles Coated with Carbon Powder under Pressurized Conditions"

_molecules, 2020, doi:10.3390/molecules25081810_

Round 1

Reviewer 1 Report

Dear Authors,

Specific comments are in the attached word document. The comments are marked after every page. There are a few general remarks for the authors to improve the quality of the manuscript. 

  1. The research question and the research methodology need to be explained in further detail. 
  2. Include a section about the flow of the paper in the introduction section, so that the readers could navigate through different sections. 
  3. Include actual pictures of the experimental set-up.
  4. Indicate the composition of the reducing gas. Further reading about the circored process is recommended, which uses fluidized bed reactors with hydrogen gas for iron-ore reduction.
  5. The manuscript has very few references and a lot of information is repeated. 
  6. The language of the article needs improvement and thorough proof-reading is required to improve the quality of the manuscript. 
  7. Some of the tables containing the details about variation of design parameters could be presented as annexures. 

Author Response

Dear editors and reviewers,

I am very grateful to your comments for the manuscript. Here below is our description on revision according to the reviewer’s comments and the changes in the manuscript within the document have been tracked by using the red colored text. 

1) Reviewer #1“The title could be modified to clearly indicate the objective of the research article. It could be "Experimental investigation of fluidization reduction characteristics of iron particles coated with carbon powder under pressurised conditions ".”

Response: 1. The title of “Effects of carbon powder coating and pressurization on the fluidization reduction of fine iron ore particles” was deleted in page 1 line 2-4.

The title of “Experimental investigation of fluidization reduction characteristics of iron particles coated with carbon powder under pressurised conditions” was added in page 1 line 2-4.

2) Reviewer #1“”

Response: 1. The word of “better operating parameters” was deleted in page 1 line 15.

The word of “optimal operating conditions” was added in page 1 line 15.

3) Reviewer #1“”

Response: 1. The word of “clarify” was deleted in page 1 line 15.

The word of “investigate” was added in page 1 line 15.

4) Reviewer #1“”

Response: 1. The word of “best operating parameter” was deleted in page 1 line 18.

The word of “optimal operating conditions” was added in page 1 line 18.

5) Reviewer #1“Rephrase the sentence.”

Response: 1. The sentence of “The optimal operating conditions for pure hydrogen to reduce Brazilian fine iron ore was found to be a reduction temperature of 923-1023 K, the linear velocity of the reducing gas was 0.6 m/s, the reducing time was 30-50 min, the reducing pressure was 0.25 MPa, the mass content of the coated carbon powder was 2%-6% (accounting for the mass of the mineral powder), and the particle size of the carbon powder was 4-7 µm. ”was added in page 1 line 18-22.

6) Reviewer #1“Optimal”

Response: 1. The word of “the best operating parameter” was deleted in page 1 line 18.

The word of “The optimal operating conditions” was added in page 1 line 18.

7) Reviewer #1“Does the author refer to Brazillian iron ore coated with carbon under pressurized conditions ? The optimal temperature for hydrogen reduction has been found to be 1073 K by many other researchers like Ranjani da costa”

Response: 1. Yes, this paper mainly studies the reduction of Brazilian iron ore powder by coating carbon powder under pressure. Under different experimental equipment and experimental conditions, the optimal reduction temperature is different. Pressurization will promote the reduction reaction and reduce the reduction temperature.

8) Reviewer #1“By adhession of particles, does the author refer to sintering”

Response: 1.NO,It's not sintering。It refers to the phenomenon that ore powder particles stick together in the process of fluidization reduction, not sintering in blast furnace ironmaking

9) Reviewer #1“More than 80% of the primary iron is produced from the blast furnace process. How has the progress of blast furnace been restricted ?”

Response: 1. The sentence of “Although blast furnace iron-making currently occupies the dominant position in the smelting of iron and steel, its reliance on metallurgical coke and emissions of pollutants, such as those generated during sintering and pelletizing, have restricted its development” was deleted in page 1 line 29-31.

The sentence of “Although blast furnace iron-making currently occupies the dominant position in the smelting of iron and steel, its reliance on metallurgical coke and emissions of pollutants, such as those generated during sintering and pelletizing, cause harm to human body and environment” was added in page 1 line 29-31.

10) Reviewer #1“Metallurgical workers ???

Response: 1. The word of “Metallurgical workers” was deleted in page 1 line 32.

The word of “metallurgical researchers” was added in page 1 line 32.

11) Reviewer #1“How does fluidization reduction compete with blast furnace or direct reduction of iron ore with syn gas in terms of cost ?

Response: 1. At present, the cost of all kinds of fluidization ironmaking technology is relatively high, and the advantage of short process is weakened, so the key to popularize this technology is to reduce the cost of inhibiting the agglomeration of ore powder and recycle the reducing gas.

12) Reviewer #1“What is the fluidization effect ?

Response: 1. The sentence of “Fang and Li [15-16] put forward the method of the fluidized bed reduction of iron ore fines, which was found to effectively improve the reduction effect of the fluidized bed.” was deleted in page 2 line 48-50.

The sentence of “Fang and Li [15-16] proposed to use the method of mixed fluidized bed of ore and coal to carry out the fluidization reduction experiment of iron ore powder, which reduced the adhesion ratio of ore powder after reduction” was added in page 2 line 48-50.

13) Reviewer #1“The authors could refer to more literature to make the list more exhaustive.?

Response: 1. References increased to 38

14) Reviewer #1“What are the adhession methods ? Are the authors referring to adhession of iron ore particles during the rediction process ?

Response: 1. Yes,the adhession methods are the adhession of iron ore particles during the rediction process

The sentence of “Summarizing previous studies, the addition of inert inhibitors is the most effective method of inhibition.” was deleted in page 2 line 53-54.

The sentence of “Summarizing previous studies, the addition of inert inhibitors is the most effective method of inhibition particle adhesion.” was added in page 2 line 53-54.

15) Reviewer #1“It is suggested that the section be titled as " Experimental methodology" or just "Methodology".

Response: 1. The word of “Experiment” was deleted in page 2 line 64.

The word of “Experimental methodology” was added in page 2 line 64.

16) Reviewer #1“Why was carbon powder selected.

Response: 1. Different sizes of carbon powder have different binding force with mineral powder. In our experiment, we choose carbon powder with similar size with mineral powder, and carbon powder far smaller than the size of mineral powder for experiment to observe the reduction effect.

17) Reviewer #1“A picture of the actual experimental set-up will add more value to the research article's quality.”

Response: 1 The sentence of “A high-pressure visible fluidized bed was selected for this experiment, and its experimental setup and experimental device diagram are shown in Figure 1 and Figure 2.” was added in page 3 line 74-75.

The picture of“

Figure 2. Experimental device diagram

” was added in page 4 line 95-96.

18) Reviewer #1“Composition of the reducing gas is not very clear. In the introduction part it was mentioned that pure hydrogen will be used as the reducing gas. In the experimental set-up a CO gas cyllinder is being used as well. It would be good to include the gas composition for which the tests have been conducted.”

Response: 1 The experimental device is equipped with multiple gas bottles, which can be mixed in a mixed gas preheating chamber to obtain the required mixed gas, but this experiment does not require mixing, only pure hydrogen is used for reduction.

19) Reviewer #1“Why does the temperature start rising simultaneously ? How is the heat being added to the system ?”

Response: 1 The experiment uses a shaft furnace. The furnace is heated by electric heating elements in the furnace, and the temperature in the furnace is observed and adjusted by thermocouples.

20)Reviewer #1“indicate the design temperature.”

Response: 1 The sentence of “When the reduction temperature reached the design temperature (1023 K) the N2 valve was closed and the H2 valve was opened to begin the experiment.” was added in page 5 line 111-112.

The sentence of “Cooling to room temperature, the bonded and unbonded mineral powder in the fluidized bed were taken out, and the sticking ratio was calculated.” was added in page 5 line 111-112.

21)Reviewer #1“At what temperature is the hydrogen gas entering the reduction chamber ? is it mixed with carbon monoxide as well ?

Response: 1 The temperature of hydrogen entering the reduction chamber is 1023K, without mixing with carbon monoxide

22)Reviewer #1“Provide suitable references for the two methods.?

Response: 1. The word of “[22-25]” was added in page 5 line 123.

The sentence of “[22] B. Zhang, Z. Wang, X. Z. Gong, et al. Influence of Fluidization Conditions on Sticking Time During Reduction of Fe2O3 Particles with CO[J]. Journal of Pharmaceutical Sciences, 2012, 59(8):1174–1177.

[23] L. Guo, H. Gao, J. T. Yu, et al. Influence of hydrogen concentration on Fe2O3 particle reduction in fluidized beds under constant drag force[J]. International Journal of Minerals Metallurgy & Materials, 2015, 22(1):12-20.

[24] L. Guo, J. T. Yu, J. K. Tang, et al. Influence of Coating MgO on Sticking and Functional Mechanism during Fluidized Bed Reduction of Vanadium Titano-magnetite[J]. Journal of Iron & Steel Research International, 2015, 22(6):464-472.

[25] L. Guo, J. Tang, H. Tang, et al. Influence of Different MgO Coating Methods on Preventing Sticking During Reduction of Fe2O3 Particles in a Fluidized Bed[J]. Materials Today: Proceedings, 2015, 2:S332-S341.” was added.

23)Reviewer #1“What is bonded and unbonded mineral powder? Either explain the terms or provide suitable references.?

Response: 1. The word of “[20-21]” was added in page 5 line 111.

The sentence of “[20] Q. Y. Xu, Z. P. Li, Z. Z. Liu, et al. The Effect of Pressurized Decarbonization of CO onInhibiting the Adhesion of Fine Iron Ore Particles. Metals, 2018, 8(7): 525.

[21]E. Hamed, Y. Frostig. Natural frequencies of bonded and unbonded prestressed beams–prestress force effects[J]. Journal of Sound & Vibration, 295(1-2):28-39.

” was added.

24)Reviewer #1“It is not clear, what the different schemes selected by the authors are and why they were chosen ?”

Response: 1. The pre-experiment is to prove that pressurization and carbon powder can improve the fluidization experiment. The orthogonal experiment is to specifically study the influence of various factors on the reduction experiment.

25)Reviewer #1“Provide adequate reference to design of experiments to ensure that the theoretical basis of the experiments is clear to the reader.?”

Response: 1. The word of “[26-28]” was added in page 6 line 153.

The sentence of “[26] D. S. Sui, Z. S. Cui. Application of orthogonal experimental design and Tikhonov regularization method for the identification of parameters in the casting solidification process[J]. Acta Metallurgica Sinica(English Letters), 2009, 221(1):13-21.

[27] Y. G. Chen, W. M. Ye, K. N. Zhang. Strength of copolymer grouting material based on orthogonal experiment[J]. Journal of Central South University, 16(1):143-148.

[28] X. Zheng, Y. C. Ye, F. Y. Liu. Study of the Pure Tantalum Plate Process by Means of an Orthogonal Experimental Design[J]. 1998, 27(3):173-176.

” was added.

26)Reviewer #1“Can the results be presented in the form of a graph or a visuallization ?”

Response: 1. The graphical form of the experimental results is shown in Figure 3.

27) Reviewer #1“Is that the title of a new section ?

Response: 1 Yes, The above is the analysis of experimental data results, this is a chemical reaction analysis

28) Reviewer #1“Unreacted shrinking core model ?

Response: 1 Yes, The unreacted core model of mineral powder has been widely used because of its simple mathematical treatment and the results close to the actual situation. Because with the reduction reaction going on, the reaction interface continues to move towards the inner part of the ore ball, and the unreacted core radius continues to decrease, leading to the tendency of the ball shrinking, it is called Unreacted shrinking core model.

The sentence of “unreacted nuclear model” was deleted.

The sentence of “Unreacted shrinking core model” was added in page 13 line 280, page 14 line 287, page 14 line 298, page 16 line 337.

29) Reviewer #1“A better indicator would be mass flow rate of the gas to understand how much hydrogen gas is required to reduce iron or particles and how does the amount get affected by addition of carbon particles.

Response: 1 Thank you editor for reminding me that I will use the mass flow meter for experiments in the future.

30) Reviewer #1“Provide suitable reference for Taman temperature.”

Response: 1. The word of “[37-38]” was added in page 18 line 397.

The sentence of“

 [37]R.T.K. Baker. The relationship between particle motion on a graphite surface and Tammann temperature[J]. 78(2):473-476.

[38] L. J. Zhou, W. L. Wang. Study of the Viscosity of Mold Flux Based on the Vogel–Fulcher–Tammann (VFT) Model[J]. Metallurgical & Materials Transactions B, 2016, 48(1):1-7.

 was added

Thank you for the kind advice.

Sincerely yours,

Dr. Xu

Reviewer 2 Report

Dear Editor,

I have gone through the manuscript titled "Effects of carbon powder coating and pressurization on the fluidization reduction of fine iron ore particles". The manuscript includes useful information since it addresses an industrial complication. However, the conclusions and discussion should be improved. There are missing information. I inserted my 17 comments in the attached pdf file.

The manuscript in its current form is not suitable for publication in your journal. Therefore, I only recommend a "major revision".

If the authors fixed the raised issues, I would be more than happy to review the manuscript again.

Kind regards

Author Response

Dear editors and reviewers,

I am very grateful to your comments for the manuscript. Here below is our description on revision according to the reviewer’s comments and the changes in the manuscript within the document have been tracked by using the red colored text. 

1) Reviewer #2  “”

Response: The word of under. was deleted in page1 line 15.

2) Reviewer #2  “What about the carbon characteristics? Recent studies showed that the nature of carbon plays a key role in the reduction kinetics (Journal of Cleaner Production 219,971-980(2019); Materials. Metall and Materi Trans B 48,2316-2323(2017)). These characteristics include carbon texture (i.e.. Surface area and porosity) and degree of graphitization.”

Response: The sentence of “Mohannad and Maroufi [13-14] believe that the properties of the toner (the structure of the toner and the degree of graphitization) play an important role in the reduction kinetics.” was added in page 2 line 46-47.

3) Reviewer #2  “Not clear statement. Where and how this may occur?”

Response: The sentence of “in the process of high temperature fluidization reduction” was added in page 2 line 37-38

4) Reviewer #2  “

Response: The word of both domestically and abroad was deleted in page 2 line 41.

5) Reviewer #2  “Bit confusing. Maybe silica and calcium oxide?”

Response: The word of silica calcium oxide was deleted in page 2 line 44.

The word of silica and calcium oxide was added in page 2 line 44.

6) Reviewer #2  “The technical term “adhesion loss” should be explained?”

Response: The word of adhesion loss” was deleted in page 2 line 53.

The word of adhesion of iron ore powder was added in page 2 line 53.

7) Reviewer #2  “What do you mean by total iron? In which form iron is present? If these analysis was obtained by XRF, iron is usually reported as iron in the oxide in the forms of hematite or wusite. ”

Response: The sentence of “Brazilian iron ore powder is mainly composed of hematite, TFe refers to the total iron content determined by chemical analysis of iron ore samples” was added in page 3 line 67-68.

8) Reviewer #2  “The sulfur may impact the quality of steel. Is the percentage here within the practical range for iron ore reduction? ”

Response: the part of “Table 2. Chemical composition of toner (mass %).

Fixed carbon

Ash

Volatile matter

Sulfur content

Phosphorus content

Moisture

>82

<15

<3

<0.8

<0.015

<5

dad been deleted in page 3 line72.

the part of “Table 2. Chemical composition of toner (mass %).

Fixed carbon

Ash

Volatile matter

Phosphorus content

Moisture

>82

<15

<3

<0.015

<5

”was added in page 3 line72.

9) Reviewer #2  “Typo ”

Response:Extra spaces are removed in page 3 line 75.

10) Reviewer #2  “Typo ”

Response: The word of tore powder was weighed to 20 g was deleted in page 4 line 100.

The word of 20 g of iron ore powder was added in page 4 line 100.

12) Reviewer #2  “How these two parameters were assessed? ”

Response: The sentence of “The higher the metallization rate, the better the quality of the ore powder after reduction. The lower the adhesion ratio, the better the fluidization state during the reduction process.” was added in page 5 line 119-121.

13) Reviewer #2  “What does the “metallization ratemean? I assure it means “reduction rateor “conversion rate”. I suggest the authors to use commonly known technical expression ? ”

Response: The sentence of “Metallization rate refers to the ratio of the metal iron content to the total iron content. The metal iron should include zero-valent iron and the amount of iron combined with carbon. Total iron refers to the total iron content determined by chemical analysis of iron ore samples.” was added in page 5 line 117-119.

14) Reviewer #2  “The x-axis should be defined and made legible? ”

Response: According to the reviewer#2's opinion, the part of “

”was added in page 10 line 187

16) Reviewer #2  “Most references here are outdated. Up-to-date reference should be added. ”

 Response: References increased to 38

11,15,17) Reviewer #2  “”

Response: According to the reviewer#2's opinion, the part of “

 During the carbon coating reduction process, carbon participates in the reaction to reduce iron oxide. The chemical reaction is as follows:

            (17)

              (18)

                      (19)

              (20)

The reduction is also enhanced by iron oxide reduction by carbon. This reduction occurs by direct solid-solid reaction which produces carbon monoxide and carbon dioxide. Carbon dioxide again attacks carbon to form carbon monoxide (Boudouard reaction) and to form very reactive porous carbon [34]. Porous carbon enhances the reduction of iron oxide [35], while Carbon monoxide is very good reductant for most metal oxide [36]. Therefore, the addition of carbon powder can improve the reduction effect of iron oxide.

”was added in page 10 line 375-386

Thank you and all the reviewers for the kind advice.

Sincerely yours,

Dr. Xu

Reviewer 3 Report

Please refer to the attached pdf that has the report + the marked manuscript.

Author Response

Dear editors and reviewers,

Thank you for your valuable comments. Your statistical analysis and regression model are very meaningful for the experimental research. However, in view of the comments of editors 1 and 2, I keep the structure and content of the paper unchanged. I will apply the data analysis method you proposed to the future papers

Here below is our description on revision according to the reviewer’s comments and the changes in the manuscript within the document have been tracked by using the red colored text. 

1) Reviewer #3“”

Response: 1. The word of “was the main reason” was deleted in page 1 line 23.

The word of “were the main reasons” was added in page 1 line 23.

2) Reviewer #3“not clear what this mean.”

Response: 1. The sentence of “To improve the fluidization effect” was deleted in page 1 line 41.

The sentence of “To reduce the sticking ratio and increase the metallization rate” was added in page 2 line 41.

3) Reviewer #3“mole fraction?”

Response: 1. The word of “specific gravity of CO” was deleted in page 2 line 52.

The word of “gas mass fraction of carbon monoxide” was added in page 2 line 52.

4) Reviewer #3“the different quality, effect?

Response: 1. The word of “the different quality, the particle size” was deleted in page 2 line 59.

The word of “the quality and particle size of toner” was added in page 2 line 59.

Response: 2 “Reduction effect ” refers to the sticking ratio and metallization rate of iron ore powder after reduction

5) Reviewer #3“Some modifications in page 3

Response: 1. The sentence of “, and its composition is presented” was deleted in page 3 line 68.

Response: 2. The sentence of “, and its composition is presented” was deleted in page 3 line 70.

Response: 3. The word of “visible” was deleted in page 3 line 74.

The word of “transparent” was added in page 3 line 74.

Response: 4. The sentence of “, and the schematic diagram is presented” was deleted in page 3 line 75.

Response: 5. The word of “gas distribution plate” was deleted in page 3 line 77.

The word of “Gas mixing hole” was added in page 3 line 76.

Response: 6. The word of “meters” was deleted in page 3 line 79.

The word of “controllers” was added in page 3 line 79.

Response: 7. The word of “can be” was deleted in page 3 line 82.

The word of “is” was added in page 3 line 82.

6) Reviewer #3“Some modifications in page 4

Response: 1. The sentence of “iron ore powder was weighed to 20 g for each experiment and added to the tube.” was deleted in page 4 line 100-101.

The sentence of “20 g of iron ore powder for each experiment was added to the tube.” was added in page 4 line 100-101.

Response: 2. The sentence of “The experimental conditions and results are presented” was deleted in page 4 line103-104

Response: 3. The word of “first” was deleted in page 4 line 104.

Response: 4. The sentence of “The temperature began to rise simultaneously.” was deleted in page 4 line 106.

The sentence of “Then the fluidized bed starts to heat up” was added in page 4 line 106.

7) Reviewer #3“agglomerated?

Response: 1.Yes, Iron ore fines are bound in the form of agglomerates

8) Reviewer #3“”

Response: 1. The sentence of “

Additionally, three levels were considered for each factor. A, B, C, D, E, and F respectively represent the six factors, where A is the reduction temperature, A1 is 973 K, A2 is 1023 K, and A3 is 1073 K; B is the linear velocity of the reducing gas, B1 is 0.4 m/s, B2 is 0.6 m/s, and B3 is 0.8 m/s; C is the reduction time, C1 is 30 min, C2 is 50 min, and C3 is 70 min; D is the reduction pressure, D1 is 0.1 MPa, D2 is 0.25 MPa, and D3 is 0.4 MPa; E is the particle size of the toner, E1 is 4-7 µm, E2 is 48-75 µm, and E3 is 150-180 µm; finally, F is the mass content of the toner (accounting for the mass ratio of the reduced mineral powder), F1 is 2%, F2 is 4%, and F3 is 6%. The level of each factor is presented in Table 4, and the orthogonal experiment plan[26-28] is described in Table 5. 

 was deleted in page 6 line 151-153.

The sentence of “Additionally, three levels were considered for each factor. A, B, C, D, E, and F respectively represent the six factors in (Table 4), and the orthogonal experiment plan[26-28] is described in (Table 5).” was added in page 6 line 151-153.

9) Reviewer #3“”

Response: 1. The sentence of “In each experiment, 20 g of fine iron ore particles and a certain size and proportion of carbon powder were mixed. Before the experiment, the fluidized bed was preheated to 873 K, and the air in the bed was discharged after introducing N2. The temperature of the fluidized bed was then set to the reduction temperature, and pure H2 was introduced to reduce the fine iron ore particles. The reduction time was arranged according to the setting. After the reduction,” was deleted in page 7 line 157.

10) Reviewer #3“”

Response: 1.The sentence of “Tables 7 and 8 present the analysis results of the orthogonal experimental data.” was deleted in page 9 line 173.

11) Reviewer #3“”

Response: 1.The sentence of “From the experimental results, it can be seen that.” was deleted in page 12 line 245.

12) Reviewer #3“”

Response: 1 The sentence of “

3Fe2O3(s)+H2(g)=2Fe3O4(s)+H2O(g)     ΔG=-41425-111.42T               ï¼ˆ1)

Fe3O4(s)+H2(g)=3FeO(s)+H2O(g)       ΔG=66105-69.16T                 ï¼ˆ2)

FeO(s)+H2(s)=Fe(s)+H2O(g)        ΔG=17580-11.60T               (3)

Fe3O4(s)+4H2(g)=3Fe(s)+4H2O(g)      ΔG=29700-25.4T                  (4)

” was deleted in page 13 line 280-283.

The sentence of “

3Fe2O3(s)+H2(g)=2Fe3O4(s)+H2O(g)     ΔG=-41425-111.42T               ï¼ˆ1)

Fe3O4(s)+H2(g)=3FeO(s)+H2O(g)       ΔG=66105-69.16T                 ï¼ˆ2)

FeO(s)+H2(s)=Fe(s)+H2O(g)         ΔG=17580-11.60T                    ï¼ˆ3)

Fe3O4(s)+4H2(g)=3Fe(s)+4H2O(g)      ΔG=29700-25.4T                  (4)

.” was added in page 13 line 269-272.

13) Reviewer #3“”

Response: 1 The sentence of “As shown in Figure 4, as the reduction reaction progresses, a metallic Fe product layer is gradually formed, and the core radius of unreacted Fe2O3 is gradually reduced.” was deleted in page 14 line 288-289.

The sentence of “As the reduction reaction progresses, a metallic Fe product layer is gradually formed, and the core radius of unreacted Fe2O3 is gradually reduced in (Figure 5).” was added in page 14 line 288-289.

14) Reviewer #3“”

Response: 1 The sentence of “The reducing gas (H2) is diffused through the gas film to the surface of the solid product layer (Fe).” was deleted in page 14 line 291-292.

The sentence of “The reducing gas (H2) diffuses through the gas film to the surface of the solid product layer (Fe).” was added in page 14 line 291-292.

15) Reviewer #3“?”

Response: 1 The sentence of “An H2 reducing gas with a linear velocity of 0.6 m/s was passed to 20 g of Brazilian fine iron ore particles for fluidized reduction” was deleted in page 15 line 312-313.

The sentence of “An H2 reducing gas with a linear velocity of 0.6 m/s passes 20 g of Brazilian fine iron ore particles for fluidized reduction” was added in page 15 line 312-313.

16) Reviewer #3“?”

Response: 1. The word of “Se” was deleted in page 16 line 342.

The word of “Sc” was added in page 16 line 342.

17) Reviewer #3 “The authors have got to do a better job citing literature. Engineering articles reference 30 to 40 articles”

Response: 1 References increased to 38

Thank you and all the reviewers for the kind advice.

Sincerely yours,

Dr. Xu

Round 2

Reviewer 1 Report

Dear Authors,

Thanks for making changes to the original manuscript. I have some comments. I have divided them into specific comments and general comments. 

Specific comments :

  1. In the Introduction section, a paragraph describing the flow of the research article will improve the readability of the article and allow readers to jump from one section to the other based on their specific interests.
  2. The section on experimental methodology could be renamed methodology and the experimental methodology and "Design of experiment" methodology could be presented as sub-sections. Providing the entire methodology in one section could improve the repeatability of the experiments by other researchers. 
  3. The methodology section could be followed by the results and discussion section, where  results from the experiments and the factoral analysis are presented. A brief discussion on the results could be presented in this section or a separate section on discussions could be included.
  4. The final section could be the conclusion section. A sub-section or few ideas about future work could be included in this section. 
  5. Figure 2 : It is unclear what A and B represent. If possible, a larger image of the experimental set-up could be included indicating the different components and their make, so that other researchers can replicate the results. 

General comments 

  1. While the article is focussed on fluidized bed reactors for reducing iron oxide with pure hydrogen as an alternative to traditional blast furnace technology, it could be a good idea to briefly mention other alternative technologies like direct reduction of iron ore with hydrogen, BF-CCS, electrolysis of molten iron ore etc. 
  2. Further proof-reading is required to improve the quality of the article.

Author Response

Dear editors and reviewers,

I am very grateful to your comments for the manuscript. Here below is our description on revision according to the reviewer’s comments and the changes in the manuscript within the document have been tracked by using the red colored text. 

1) Reviewer #1“a.In the Introduction section, a paragraph describing the flow of the research article will improve the readability of the article and allow readers to jump from one section to the other based on their specific interests. ".”

Response:1.the paragraph of “To investigate the effect of carbon coating on the inhibition of the adhesion of iron ore powder particle, a six-factor, three-level orthogonal experimental method was used on a high-pressure visible fluidized bed experimental device to study the reduction temperature, the reduction time, the gas linear velocity, the reduction pressures, the quality and particle size of toner on the reduction effect, to obtain the best operating parameters. Based on the subsequent analysis, the mechanism of the bonding of fine iron ore particles after reduction and the mechanism of carbon powder coating to suppress the bonding are analyzed to provide data and a theoretical basis for the fluidized reduction iron-making process.” was deleted in page 2 line 52-60.

the paragraph of “In this paper, Brazil iron ore powder is used as the experimental material to conduct the experiment of reducing iron ore powder in the high-pressure visible fluidized bed. First, four sets of preliminary experiments were conducted to analyze the effects of pressure and coated carbon powder on the fluidized reduction of Brazilian iron ore powder. Then select 6 influencing factors to design 27 sets of orthogonal experiments, determine the primary and secondary factors affecting the fluidization reduction effect of Brazilian iron ore powder through the experimental results, and obtain the best operating parameters through comprehensive analysis. Finally, the mechanism of iron ore powder particle adhesion and the mechanism of coating carbon powder to inhibit iron ore powder adhesion are analyzed, which is the data and theoretical support for fluidized ironmaking process.” was added in page 2 line 52-60.

2) Reviewer #1“b. The section on experimental methodology could be renamed methodology and the experimental methodology and "Design of experiment" methodology could be presented as sub-sections. Providing the entire methodology in one section could improve the repeatability of the experiments by other researchers. ".”

Response: 1. The second section of the article was renamed ‘Methodology’, Section 2.2 was renamed ‘Experimental Instruments and Methods’, Section 2.3 was added and renamed ‘Experimental Design’.

For the complete experimental method, see section ‘2.2 Experimental Instruments and Methods ’of the article

3) Reviewer #1“c. The methodology section could be followed by the results and discussion section, where results from the experiments and the factoral analysis are presented. A brief discussion on the results could be presented in this section or a separate section on discussions could be included.

Response: 1.Sections 3 and 4 are merged into Section 3, renamed ‘The experimental results and discussion’, and analysis and discussion of experimental data

4) Reviewer #1“d. The final section could be the conclusion section. A sub-section or few ideas about future work could be included in this section.”

Response: 1. The sentence of “In the future, the experiment is planned in two directions. On the one hand, the fluidization experiment will be continued. A flue gas analyzer will be installed at the gas outlet to determine the degree of reduction of iron ore powder in the fluidized bed by analyzing the composition of the exhaust gas. On the other hand, the reduced iron ore powder is subjected to hot briquetting treatment, and then put into an electric furnace as a steelmaking raw material to smelt special steel. ’was added in page 18 line 390-394.

5) Reviewer #1“e.Figure 2 : It is unclear what A and B represent. If possible, a larger image of the experimental set-up could be included indicating the different components and their make, so that other researchers can replicate the results.”

Response: 1. The picture of“

Figure 2. Experimental device diagram

” was deleted in page 4 line 88-89.

The picture of“

Figure 2. Experimental device diagram

” was added in page 4 line 88-89.

Thank you and all the reviewers for the kind advice.

Sincerely yours,

Dr. Xu

Reviewer 2 Report

Dear Editor,

I acknowledge the authors to answer all the comments. The issues raised by the reviewers have been addressed apropriately. Therefore, I recommend its publication in your journal.

Kind regards

Author Response

Dear editors and reviewers,

I am very grateful to your comments for the manuscript. Thank you very much for your support.

Sincerely yours,

Dr. Xu

Reviewer 3 Report

There are some nagging English grammar issues that could improve how the paper reads.  I still think that there is a bit of hand waving with respect to how the factors affect the primary variables.  The authors should pay attention to how they report their data and should apply SI notation (https://doi.org/10.1002/cjce.22534) as well as significant figures.  There is no way they can justify 5 significant figures on G (Equations 1-4), or 3 and 4significant figures on most of the other data (55.12%, 67.12%, 79.86%, 87.05%, 88.80%, 88.96%, and 86.81%).  Please look at https://doi.org/10.1002/cjce.22532

https://doi.org/10.1002/cjce.22509

Finally, for fluidization papers, the authors must include a table with the physical properties pertinent to fluidization: see Figure 1 of  https://doi.org/10.1002/cjce.23517 to get an idea of what they need to add.  Otherwise, people in fluidization really won't get what is going on.

Author Response

Dear editors and reviewers,

I am very grateful to your comments for the manuscript. Here below is our description on revision according to the reviewer’s comments and the changes in the manuscript within the document have been tracked by using the red colored text. 

1) Response: 3.The format in the paper is modified according to BIPM standard

2)Response: 3.the paragraph of “With H2 as a reducing agent, the reactions that occur during the reduction of iron ore are as follows.

3Fe2O3(s)+H2(g)=2Fe3O4(s)+H2O(g)     ΔG=-41425-111.42T               ï¼ˆ1)

Fe3O4(s)+H2(g)=3FeO(s)+H2O(g)       ΔG=66105-69.16T                 (2)

FeO(s)+H2(s)=Fe(s)+H2O(g)         ΔG=17580-11.60T                    ï¼ˆ3)

Fe3O4(s)+4H2(g)=3Fe(s)+4H2O(g)      ΔG=29700-25.4T                  ï¼ˆ4)

” was deleted in page 12 line 246-249.

the paragraph of “With H2 as a reducing agent, the reactions that occur during the reduction of iron ore are as follows.

3Fe2O3(s)+H2(g)=2Fe3O4(s)+H2O(g)                    ï¼ˆ1)

Fe3O4(s)+H2(g)=3FeO(s)+H2O(g)                       ï¼ˆ2)

FeO(s)+H2(s)=Fe(s)+H2O(g)                          ï¼ˆ3)

Fe3O4(s)+4H2(g)=3Fe(s)+4H2O(g)                      ï¼ˆ4)

” was added in page 12 line 246-249.

3) Response: 3. The sentence of “[34] P. A. Patience, D. C. Boffito, G. S. Patience. How do you write and present research well? 11-Respect SI writing conventions[J]. Canadian Journal of Chemical Engineering, 2016, 94(8):1431-1434.

[35] D. C. Boffito, P. A. Patience, B. Srinivasan, et al. How do you write and present research well? 10-State the uncertainty, but not too precisely[J]. The Canadian Journal of Chemical Engineering, 2016.

[36] G. S. Patience, B. Srinivisan, M. Perrier, et al. How do you write and present research well? 9-show and state what error bars represent[J]. Canadian Journal of Chemical Engineering, 2016, 94(7):1221-1224.

[37] M. Menéndez, J. Herguido, A. Bérard, et al. Experimental Methods in Chemical Engineering: Reactors-fluidized Beds[J]. Canadian Journal of Chemical Engineering, 2019. ” was added.

Thank you and all the reviewers for the kind advice.

Sincerely yours,

Dr. Xu
